

**Realization of Daily Evapotranspiration in Arid Ecosystems Based on Remote Sensing**
**Techniques**
Mohamed Elhag and Jarbou A. Bahrawi
Department of Hydrology and Water Resources Management, Faculty of Meteorology,
Environment & Arid Land Agriculture, King Abdulaziz University
Jeddah, 21589. Saudi Arabia.
*Correspondence to: melhag@kau.edu.sa*
**Abstract**
Daily evapotranspiration is a major component in water resources management plans. In arid
ecosystems, the quest for efficient water budget is always hard to achieve due to insufficient
irrigational water and high evapotranspiration rates. Therefore, monitoring of daily
evapotranspiration is a keystone practice for sustainable water resources management, especially
in arid environments. Remote Sensing Techniques offered a great help to estimate the daily
evapotranspiration on a regional scale. Existing open source algorithms proved to estimate daily
evapotranspiration in arid environments comprehensively. The only deficiency of these
algorithms is course scale of the used remote sensing data. Consequently, the adequate
downscaling algorithm is a compulsory step to rationalize an effective water resources
management plans. Daily evapotranspiration was fairly estimated using AATSR in conjunction
with MERIS data acquired in July 2013 with one-kilometer spatial resolution and 3 days
temporal resolution under SEBS model. Results were validated against reference
evapotranspiration ground truth values using standardized Penman-Monteith method with $R^2$ of
0.879. The findings of the current research are successfully fulfilled to monitor turbulent heat
fluxes values estimated from AATSR and MERIS data with a temporal resolution of 3 days only
in conjunction with reliable meteorological data. Research verdicts are necessary inputs for well-
informed decision-making process regarding sustainable water resources management.

**Keywords**: Arid Environments, AATSR data, MERIS data, Remote Sensing, SEBS, Water
Resources Management.





## 1. Introduction

Evapotranspiration is the principle process in defining mass and energy relationship in the surrounding hydrosphere (Allen et al., 2007a,b; Cruz-Blanco, et al., 2014). The consumptive use of irrigational water in agriculture is the fundamental component of a balanced estimation of evapotranspiration (Bastiaanssen et al., 1998; Cammalleri and Ciraolo, 2013).

The concept of water use efficacy is basically depending on the reliable estimation of evapotranspiration and surface water evaporation (Berengena and Gavilán, 2005; Elhag et al., 2011). Weather and wind conditions induce a regional and seasonal variation of evapotranspiration estimation (Hanson, 1991; Cristobal and Anderson 2012).

Conventional techniques of field scale evapotranspiration estimations are fairly achieved epically over homogenous surfaces using ordinary techniques: lysimeter systems, Eddy Covariance (EC) and Bowen Ratio (BR). Nevertheless, conventional methods of evapotranspiration estimations are incapable of fulfilling the quest of regional evapotranspiration estimation specifically in harsh climatic conditions (Gavilán et al., 2006; Ghilain et al., 2011). Therefore, remote sensing evapotranspiration models are adequate techniques to bring satisfactory estimates (Allen et al., 2007a,b; De Bruin et al., 2010).

Remote sensing evapotranspiration models are numerous. Several algorithms are already in practice with different complexity levels to estimate fairly evapotranspiration based on different climatic conditions and land use variability (Elhag et al., 2011; Espadafor et al., 2011; Cristobal, and Anderson 2012).

Based on several scholarly work of Roerink et al., (2000); Su, (2002); Crago et al., (2005); Cha´vez et al., (2005); Loheide and Gorelick, (2005); Allen et al., (2007a,b); Ghilain et al.,





(2011); Psilovikos and Elhag (2013) on remote sensing evapotranspiration based algorithms,
there are principally two types of evapotranspiration estimation concepts on terrestrial surfaces.
The first concept is to use the surface reflectance in different visible (VIS), near-infrared (NIR)
and even extended to Thermal Infrared (TIR) portions of the electromagnetic spectrum to
rationalize the Surface Energy Balance (SEB). The other concept it to use vegetation indices
derived from canopy reflectance to conceptualize remotely sensed crop coefficient ($K_{cr}$).
Gourd truth data collection exercised at less than one-meter canopy height, in which all related
surfaces fluxes and atmospheric surface variables of the vegetation cover takes place in arid
environment takes place (Beljaars and Holtslag 1991; Zwart and Bastiaanssen, 2004). Based on
Brutsaert (1991, 1999), Monin-Obukhov Similarity (MOS) and Bulk Atmospheric Boundary
Layer (ABL) functions were calculated. Brutsaert (1999) suggested sets of criteria estimate MOS
or ABL if it scaled down appropriately for a given circumstances. Brutsaert criteria are valid
only for unstable conditions.
Therefore, van den Hurk and Holtslag (1995) adjusted and validated Brutsaert criteria using
atmospheric surface layer scaling according to Brutsaert (1982) to be used in stable conditions.
Generic estimation of Surface Albedo for vegetated land covers is based on the red (R) and NIR
band reflectance (Brest and Goward 1987) model.
The aim of the current study is to monitor turbulent heat fluxes in Wadi Ad Dawasir to estimate
the daily evapotranspiration rate and relative evaporation ratio using AATSR and MERIS
sensors. The final step is to identify the regression coefficient between the estimated
evapotranspiration's rates and the actual ground truth data.



## 2. Materials and methods

### 2.1. Study area

The study area, Wadi Ad Dawasir town is located on the plateau of Najd at Lat 44º 43' and Lon 20º 29'; about 300 km south of the capital city Riyadh. This study area comprised of gravelly tableland disconnected by insignificant sandy oases and isolated mountain bundles. Across the Arabian Peninsula as a whole, the tableland slopes toward the east from an elevation of 1,360 meters in the west to 750 meters at its easternmost limit. Wadi Ad Dawasir and Wadi ar Rummah the most important pattern of the ancient riverbeds remains in the study area. Wadi Ad Dawasir and Najran regions are the major irrigation water abstraction from Al-Wajid aquifer. Agriculture in Wadi Ad Dawasir area consists of technically highly developed farm enterprises that operate modern pivot irrigation system. The size of center pivot ranges from 30 ha to 60 ha with farms managing hundreds of them with the corresponding number of wells. The main crop grown in winter is wheat and occasionally potatoes, tomatoes or melons. All year fodder consists of alfalfa, which is cut up to 10 times a year for food. Typical summer crops for fodder are sorghum and Rhodes grass, which is perennial, but dormant in winter. The shallow alluvial aquifers could not sustain the high groundwater abstraction rates for a long time and groundwater level declined dramatically in most areas. Meteorological features of the area are speckled. Five elements of meteorology are constantly recorded through fixed weather station located within the study area. Temperature varies from 6 ºC as minimum temperature to 43 ºC as maximum temperature. Relative humidity is mostly stable at 24 %. Solar radiation of average sunrise duration is generally 11 hrs/day. Average wind speed is closer to 13 km/hr and may reach up to 46 km/hr in thunderstorm incidents. Finally, mean annual rainfall is about 37.6 mm (Al-Zahrani and Baig, 2011).



**Figure 1. Location of the study area (Elhag, 2016).**
**2.2. Methodological framework**
The current research work is based on assessing a regression correlation between estimated
evapotranspiration data conducted from AATSR and MERIS sensors and its corresponding
ground truth evapotranspiration data conducted through standardized Penman-Monteith.
Therefore, accurate synchronization of remote sensing data bypassing and ground truth data
collection were exercised.
**2.3. SEBS Model fundamentals**
Remote sensing data acquired from Advance Along Track Scanner Radiometer (AATSR) and
Medium Spectral Resolution Imaging Spectrometer (MERIS) sensors in the $8^{th}$ of July 2013
respectively. The satellite data were georeferenced to WGS-84 datum, atmospherically corrected
using SMAC correction (Rahman and Dedieu, 1994). Several meteorological data collected from
a stationary station located within the designated study area (2004-2014, average meteorological
data).
Surface Energy Balance System was initiated by Su (2002) based on further Surface Energy
Balance Index improvements. SEBS dynamicity works for regional and local ET estimation.
Regional ET estimation uses Monin–Obukhov Similarity (MOS), Bulk Atmospheric Similarity
and thermal roughness principles. On the other hand, local ET estimation uses only Atmospheric
Surface Layer (ASL) scaling fundamentals (Brutsaert 1999; Su, 2001, Su et al., 2001). The
boundary conditions (wet and dry) are essential components in ET estimation using SEBS
model. According to the water availability limitation, $H_{dry}$ is considered to be equal to the
available energy *AE* as evaporation assumed to be "zero". Following Penman–Monteith



parameterization (Monteith 1965, 1981), wet boundary condition ($H_{wet}$) is calculated as
following:
$$H_{wet} = AE - \frac{\left(\frac{\rho_a c_p}{r_{ah}}\right)\left(e_s - \frac{e}{7}\right)}{1 + \frac{\Delta}{\gamma}}$$    (1)
Where
$e$ is the actual vapor pressure ($kP_a$),
$e_s$ is the saturation vapor pressure ($kP_a$),
$c$ is the psychrometric constant ($kP_a \, ^oC^{-1}$),
$\gamma$ is the rate of change of saturation vapor pressure with temperature ($kP_a \, ^oC^{-1}$) and
$r_{ah}$ is the bulk surface external or aerodynamic resistance ($s \, m^{-1}$) .
Consequently, evaporative fraction ($\Lambda$) and relative evaporative fraction ($\Lambda_r$) are calculated as
following per image pixel:
$$\Lambda = \frac{\lambda E}{R_n - G} = \frac{\Lambda_r \, . \lambda E_{wet}}{R_n - G}$$    (2)
$$\Lambda_r = 1 - \frac{H - H_{wet}}{H_{dry} - H_{wet}}$$    (3)
Daily evaporation is estimated based on the estimation of the evaporative fraction only when the
daily net energy ($G$) and the net radiation ($R_n$) are available. Therefore, the amplitude variation
of the diurnal energy cycle is sky clarity dependent.
$$H = (1 - \Lambda).(R_n - G)$$    (4)
$$LE = \Lambda (R_n - G)$$    (5)




$$E_{daily} = \Lambda_0^{24} \cdot \int_{daytime} \cdot \frac{R_n - G_0}{\lambda_{\rho\omega}}$$    (6)
Where
$\Lambda$ is the evaporative fraction
$\Lambda_r$ is relative evaporative fraction
$R_n$ is net radiation measured in watt per square meter,
$G$ is soil heat flux measured in watt per square meter,
$H$ is turbulent sensible heat flux measured in watt per square meter,
$\lambda E$ is turbulent latent heat flux measured in watt per square meter,
$H$ is the actual sensible heat flux and determined by the bulk atmospheric similarity approach.
$P\omega$ is the density of water measured in kilograms per cubic meter.

**2.4. Validation**
Using standardized Penman-Monteith method, 50 ground truths data collected were collected
and used to validate the implemented model. The sampling locations were consistently
distributed over the designated study area. The lysimeter technique for the estimation of daily
evapotranspiration was carried out following Liu and Wang (1999) with calibrated accuracy
equal to ± 0.025. The calibration procedure was principally based on placing double
infiltrometers (Taylor 1981).
The corrected Penman equations for estimating the daily evapotranspiration was conducted
according to Jensen et al., (1990):
$$ET_o = \left[ \frac{\Delta}{\Delta + \gamma} Rn + \frac{\gamma}{\Delta + \gamma} f(U) (e_s - e_a) \right] c$$    (7)
Where
$ET_0$ is reference evapotranspiration (mm /day),
$\Delta$ is the slope of saturation vapor pressure-temperature curve (kPa /°C),


$\gamma$ is the psychrometric constant (kPa /$^o$C),
$Rn$ is the net radiation (mm (mbar)),
c is the adjustment factor,
$f(U)$ is the wind function,
$e_s$ is the saturation vapor pressure (mbar),
$e_a$ is actual vapor pressure,

Consequently, the wind function was conducted following to Doorenbos and Pruitt (1977) as:
$$f(U) = 0.27 \left(1 + \frac{U_2}{100}\right) \tag{8}$$
Where
$U_2$ is the wind speed measured surfacely at 2 m height (km/day).

Meanwhile, $e_a$ was calculated according to Allen et al. (1998) as the following:
$$e_a = \frac{e^o(T_{min})(RH_{max}/100) + e^o(T_{max})(RH_{min}/100)}{2} \tag{9}$$
Where
$e^o(T_{min})$ is the saturation vapor pressure at a daily minimum temperature (kPa),
$e^o(T_{max})$ is the saturation vapor pressure at a daily Maximum temperature (kPa),
$RH_{max}$ is the maximum relative humidity (%),
$RH_{min}$ is minimum relative humidity (%).

Linear regression model was used to find the correlation coefficient between the estimated and
the actual evapotranspiration values. Root Mean Square Error (RMSE) was used to signify the
inequality of variance and correlation of the linear regression model (Box, 1954). The R|MSE
was calculated as following:
$$RMSE = [N^{-1} \sum_{i=1}^{N}(P_i - O_i)^2]^{0.5} \tag{10}$$
Where




$N$ is the number of observations,
$P_i$ is the predicted ET values (mm/day)
$O_i$ is the calculated ET values (mm/day)

**3. Results and Discussion**
SEBS model implementation over the designated study area results in 10 different turbulent heat
fluxes thematic maps. The histogram and the scatter plot of SEBS output thematic maps were
plotted against the daily evapotranspiration values. Daily evapotranspiration values ranging from
zero to map indicates the range of the actual evapotranspiration in the study area ranges between
zero and 6.61 mm/day. The spatial distribution of the highest evapotranspiration area is the
peripheral of the agricultural pivots as it's demonstrated in Figure 2. This could be explained by
the poor drainage system were the access of irrigational water collated at the sides of the pivots
(Zwart and Bastiaanssen, 2004; Cruz-Blanco et al., 2014).The mean actual evapotranspiration
value is almost 5 mm/day (Figure 3), which is considered a high value in such arid conditions
(Elhag et al., 2011). Such evapotranspiration value supports the hypothesis of the
mismanagement of irrigational water in Wadi Ad Dawasir. Principally under extreme dry climate
conditions, relative evaporation may reach unity (Lockwood, 1999).  The relative evaporation
thematic map; demonstrated in Figure 4, confirms high correspondence between the actual and
the potential evapotranspiration, especially in the peripheral of the agricultural pivots. Normal
distribution of the relative evaporation is demonstrated in Figure 5. Mean relative evaporation
ratio is counted for 0.91. Only within the pivots, the relative evaporation decreases to 0.45
indicating the wet condition of the agricultural land (De Bruin et al., 2010). 50 points of ground
truthing data were collected during July 2013 of daily evapotranspiration. The points were



consistently distributed over the designated study area. Daily evapotranspiration estimation was
conducted according to Liu and Wang (1999) using the Lysimeter with calibrated accuracy of ±
0.025. The actual evapotranspiration data were intersected with the estimated raster image under
GIS environment. A linear regression model with $R^2$ value of 0.83 was conducted to assess the
association between the estimated and the actually measured values (Figure 6).
Implementation of SEBS model over the designated study area proved a higher daily
evapotranspiration values than projected. Higher daily evapotranspiration values were noticed
because the sensible heat flux is the major part of the energy, while the latent heat flux is
dominating only over the agricultural area (Frey et al., 2010; Elhag 2014a,b). SEBS model
behavior could be explained by the model tendency to simulate the potential daily
evapotranspiration rather than the actual daily evapotranspiration, which is identified as the lack
of Leaf Area Index value over desert areas (Li, et al., 2009; Elhag et al., 2011). The application
of SEBS model over the designated study area showed insignificance difference than the Nile
Delta Case in term of accuracy assessment (Elhag et al., 2013).

**Figure 2. Actual daily evapotranspiration thematic map.**

**Figure 3. Normal distribution of actual daily evapotranspiration data.**

**Figure 4. Relative evaporation thematic map.**



**Figure 5. Normal distribution of relative evaporation data.**

**Figure 6. The relationship between actual and simulated daily evapotranspiration.**

**Conclusions**

Projected evapotranspiration data using SEBS model and multiple remote sensing imageries demonstrated robust association with the ground truth data. The application of the SEBS model mapped the daily evapotranspiration and evaporative fraction objectively over Wadi Ad Dwaser region. The findings of the current research will help the decision makers towards modification of the agriculture activities in similar areas, in term of conservative irrigational water regulations. The model shows consistent results in the estimation of daily evapotranspiration in Nile Delta region and in Wadi Ad Dwaser. Accordingly, SEBS model can be considered as a reliable effective tool in the estimation of daily evapotranspiration explicitly in arid environments.

**Acknowledgement**

This project was funded by the Deanship of Scientific Research (DSR) at King Abdulaziz University, Jeddah, under grant no. (G-**182-155-37**). The authors, therefore, acknowledge with thanks, DSR technical and financial support.

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





**List of figures**

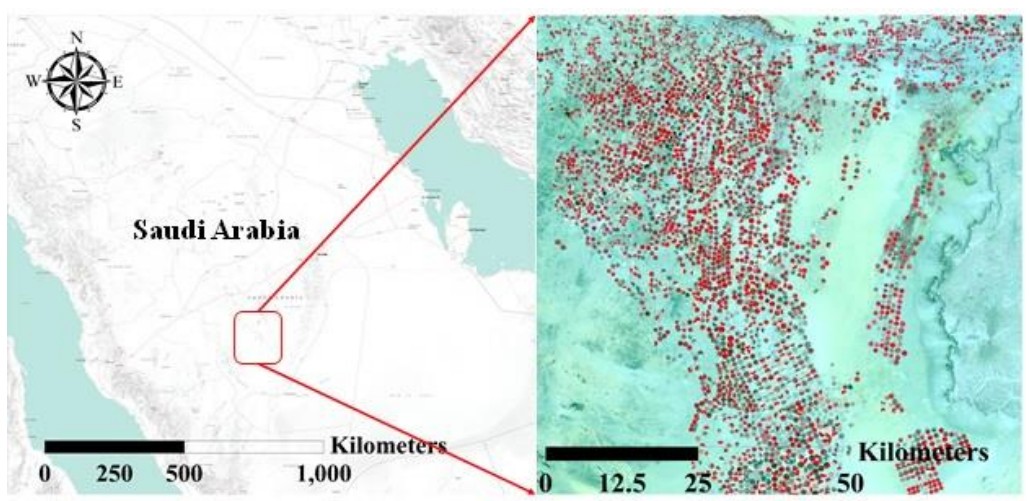


**Figure 1. Location of the study area (Elhag, 2016).**





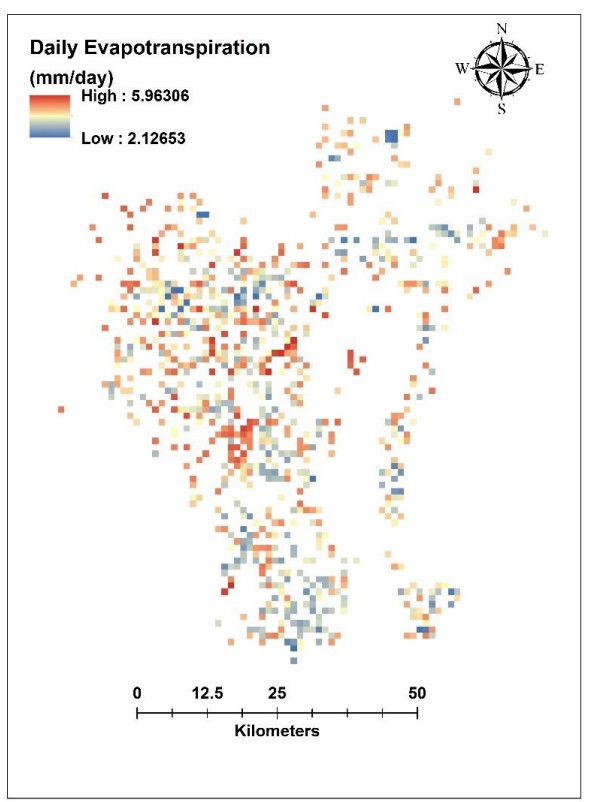


**Figure 2. Actual daily evapotranspiration thematic map.**







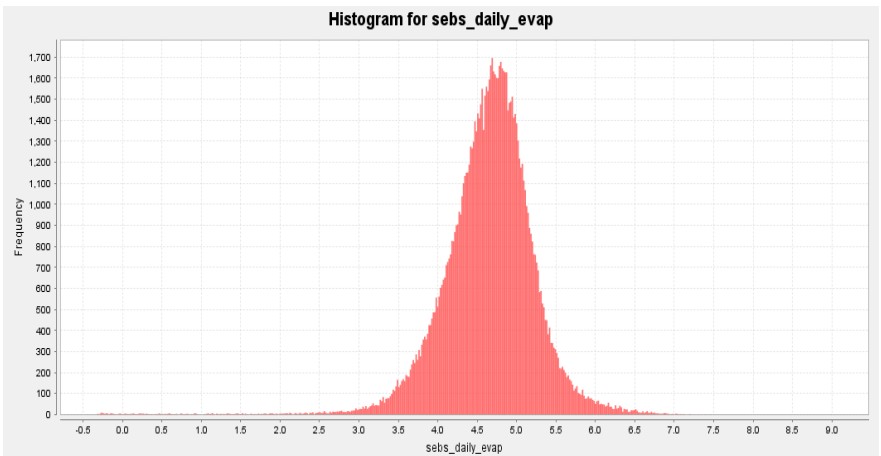


**Figure 3. Normal distribution of actual daily evapotranspiration data.**




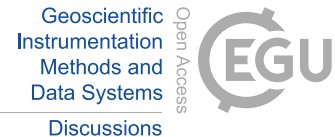



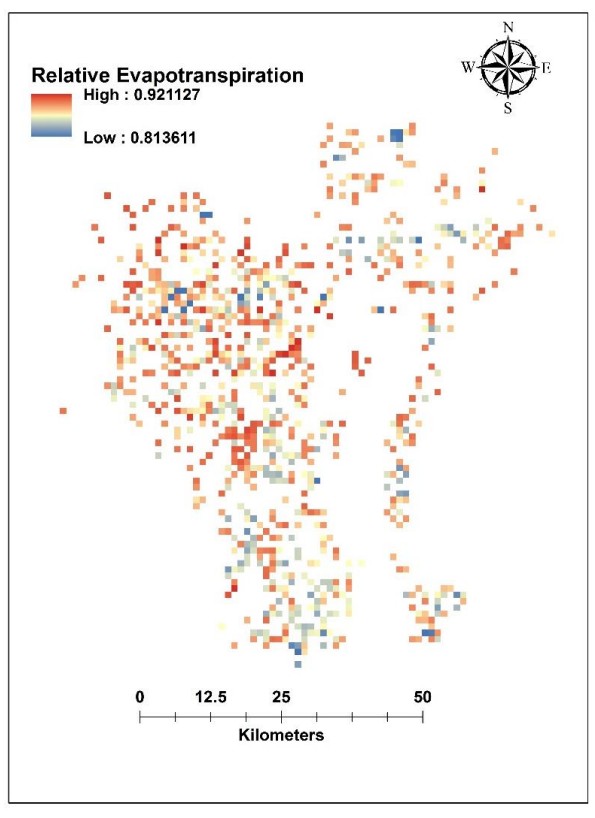


**Figure 4. Relative evaporation thematic map.**








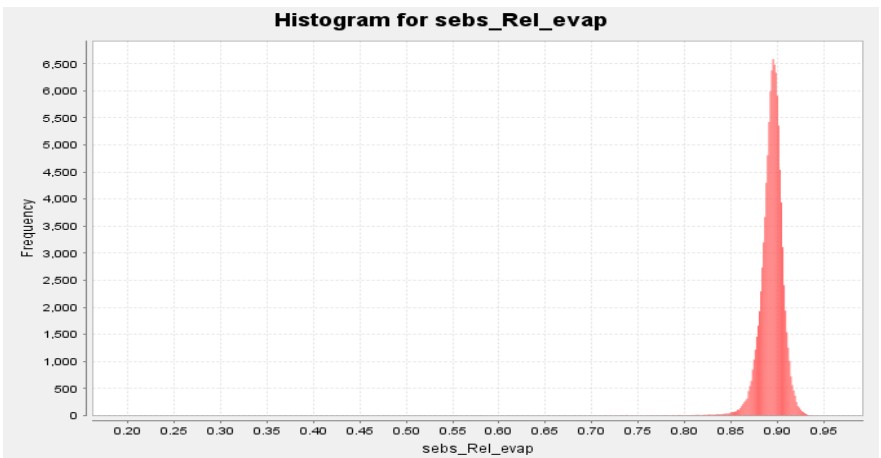


**Figure 5. Normal distribution of relative evaporation data.**










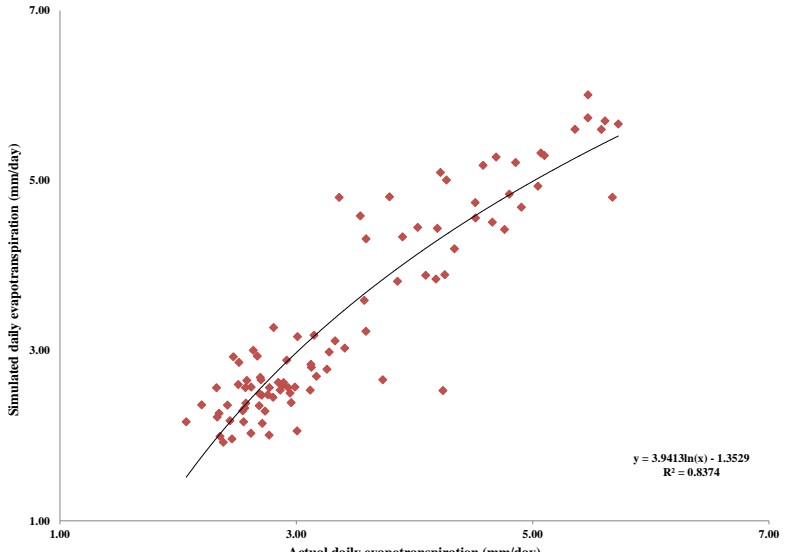


**Figure 6. The relationship between actual and simulated daily evapotranspiration.**