# Peer review of "1. Introduction"

_Geoscientific Instrumentation, Methods and Data Systems, 2016_

## Referee Comment (RC1) · S. Boteva (Referee) · 19 Dec 2016

General comments The topic of the article Realization of Daily Evapotranspiration in Arid Ecosystems Based on Remote Sensing Techniques is within the scope of Geoscientific Instrumentation, Methods and Data Systems Discussions. The title clearly reflects the contest of the paper. The aim of the research is to estimate the daily evapotranspiration rate and relative evaporation ratio using AATSR and MERIS sensors which will ease decision-making processes regarding sustainable water resources management. The paper describes the implementation of a new model and methodology in analyzing the received data. The introduction provides a generalized background of the topic, the variety of techniques used for evapotranspiration estimation

and a comparison with the remote sensing models. The abstract summarizes the described work and gives accurate information about the article. The chapter Materials and methods describes each step of the work methodology in detail along with the conducted calculations. Obtained results are well interpretated and discussed in the light of assessing the estimated and actually measured values. The authors clearly describe their original contribution in developing the new model and especially its application in arid regions. The conclusions underline the benefits for decision makers when applying the model, particularly in areas under conservative irrigational water regulations. The article is well structured with all the necessary parts for such kind of a scientific work. The references are appropriate and most of them are published during the last years.

Specific comments The language of the article and some sentences need revision. For example the sentences: "Wadi Ad Dawasir and Wadi ar Rummah the most important pattern of the ancient riverbeds remains in the study area." "Daily evapotranspiration values ranging from zero to map indicates the range of the actual evapotranspiration in the study area ranges between zero and 6.61 mm/day." There are also technical errors, for example, when figure is cited in the text it should be written Fig. not Figure. The literature cited also needs corrections in accordance with the instructions. For example: Authors wrote: Roerink, G. J., Su, Z., and Menenti, M. (2000). A 337 simple remote sensing algorithm to estimate the surface energy balance. Physics and Chemistry of the Earth, 25: 147–157. The proper way is: Roerink, G. J., Su, Z., and Menenti, M.: A 337 simple remote sensing algorithm to estimate the surface energy balance, Physics and Chemistry of the Earth, 25, 147–157, 2000. Technical corrections When mentioned for the first time, the full name of the abbreviation should be given in brackets. For example MERIS, SEBS and AATSR full names should be given in the abstract in brackets.

---

## Author Comment (AC1) · 19 Dec 2016

thanks for your comments rest assured that all of your comments will be literally followed

---

## Referee Comment (RC2) · N. Yilmaz (Referee) · 6 Jan 2017

**Review-** gi-2016-40

**Dear Author,**

- I found some mistakes related to references which are used in the paper. They are below:

1. In the page 2 (line number 33) the reference was written as **Batiaanssen et al.,1998** but in the references section **Batiaanssen et al.,1998b** Which is the correct one?

2. In the page 2 (line number 49)the reference was written as **Crago et al.,2005** but in the references section **Crago and Crowley,2005** Which is the correct one?

3. In the page 2 (line number 60) the reference **Hurk and Holtslag (1995)** was used in the text but it wasn't in the references section. Please check it.

4. In the page 6 (line number 119) the references **Monteith, 1965 and 1981** were used in the text but they weren't in the references section. Please check it.

5. In the page 7 (line number 152) the references **Liu and Wang, 1999** was used in the text but ıt wasn't in the references section. Please check it.

6. In the page 7 (line number 154) the references **Taylor, 1981** was used in the text but ıt wasn't in the references section. Please check it.

7. In the page 8 (line number 156) the references **Jensen et al.,1990** was used in the text but ıt wasn't in the references section. Please check it.

8. In the page 8 (line number 173) the references **Allen et al.,1998** was used in the text but ıt wasn't in the references section. Please check it.

9. In the page 10 (line number 211) the references **Liu and Wang, 1999** was used in the text but ıt wasn't in the references section. Please check it.

10. In the page 10 (line number 218) the references **Frey et al.,2010** was used in the text but ıt wasn't in the references section. Please check it.

11. In the page 10 (line number 221) the references **Li et al., 2009** was used in the text but ıt wasn't in the references section. Please check it.

12. In the page 10 (line number 223) the references **Elhag, et al.,2013** have to corrected as **Elhag et al., 2011.**

13. In the references section these references were written but they wasn't used in the text: **Bernstein et al., 2005**; **Loheide and Gorelick, 2005**; **Su et. al, 2010**; **Su et al., 2003**; **Tasumi et. al., 2006**; **Van den Hurk and Holstag, J995**. Please check them.

---

## Author Comment (AC2) · 6 Feb 2017

i a, very pleased to receive your valuable comments and i am also very pleased to inform you that your comments were literally followed